# Chili Pepper Carotenoids: Nutraceutical Properties and Mechanisms of Action

**DOI:** 10.3390/molecules25235573

**Published:** 2020-11-27

**Authors:** Maria Guadalupe Villa-Rivera, Neftalí Ochoa-Alejo

**Affiliations:** Departamento de Ingeniería Genética, Centro de Investigación y de Estudios Avanzados del Instituto Politécnico Nacional, Unidad Irapuato, Guanajuato 36824, Mexico; gvillarivera@gmail.com

**Keywords:** *Capsicum*, carotenoids, chili pepper, nutraceutical effects, antioxidant, cancer, cardiovascular disorders, anti-inflammatory, obesity

## Abstract

Chili pepper is a prominent cultivated horticultural crop that is traditionally used for food seasoning and is applied for the treatment and prevention of multiple diseases. Its beneficial health properties are due to its abundance and variety of bioactive components, such as carotenoids, capsaicinoids, and vitamins. In particular, carotenoids have important nutraceutical properties, and several studies have focused on their potential in the prevention and treatment of human diseases. In this article, we reviewed the state of knowledge of general aspects of chili pepper carotenoids (biosynthesis pathway, types and content in *Capsicum* spp., and the effects of processing on carotenoid content) and recent findings on the effects of carotenoid nutraceuticals, such as antioxidant, cancer preventive, anti-inflammatory, cardiovascular disorder preventive, and anti-obesity effects.

## 1. Introduction

Chili pepper (Genus *Capsicum*, Family Solanaceae) is an important cultivated spice crop. During 2018, ≈4.2 million tons of dry chilies and peppers and ≈36.8 million tons of green chilies and peppers were produced worldwide (FAOSTATS 2018) [1]. The *Capsicum* genus comprises 38 different species, but only *C. annuum*, *C. frutescens*, *C. chinense*, *C. baccatum*, and *C. pubescens* have been domesticated [2,3,4].

Chili pepper fruits have abundant biochemical and mineral constituents of nutritional value. Additionally, chili peppers are good sources of bioactive compounds, such as carotenoids (lutein, β-carotene, β-cryptoxanthin, zeaxanthin, violaxanthin, capsanthin and capsorubin), vitamins C and E, and phenolic compounds, such as flavonoids (quercetin, luteolin and phenolic acids) and capsaicinoids [5,6,7]. The content of these bioactive compounds can vary considerably depending on the chili pepper cultivar and genotype [8].

*Capsicum* fruits have been used traditionally as flavoring agents and appetite stimulators, and also for the treatment of muscle pain and toothache, parasitic infections, rheumatism, wound healing, coughs and sore throat. Moreover, chili peppers also have antiseptic, antimetastatic, antifungal, antiviral, anti-inflammatory, and immunomodulatory effects, all of which are associated with their antioxidant properties [7,9].

Pungency and color are the two main characteristics of chili pepper fruits that determine their quality. *Capsicum* comprises pungent and non-pungent fruits with a yellow, orange, or red color (Figure 1) [2]. The diverse colors of mature pepper fruits result from the accumulation of different carotenoids in the pericarp [10]. Carotenoids are naturally occurring red, brown, orange, salmon and yellow pigments found in plants, microalgae, bacteria, archaea, and a few species of fungi and aphids [11,12].

Carotenoids are mostly 40-carbon molecules with conjugated double bonds. Based on their structures, they are classified as carotenes (containing carbon and hydrogen atoms) and xanthophylls (containing carbon, hydrogen, and oxygen), and, in general, they are lipophilic compounds and usually form hydrophobic micelles [13].

Carotenoids participate in important processes in plants such as photosynthesis, photomorphogenesis, photoprotection and development. They also serve as precursors for the biosynthesis of two kinds of plant hormones (abscisic acid and strigolactones) and a diverse set of apocarotenoids. Animals cannot synthesize carotenoids de novo, but they can get them from different foods as sources of antioxidants and provitamin A [14].

Because of their antioxidant, anti-inflammatory, and photoprotective properties, carotenoids have gained relevance, and diverse investigations have been focused on their ability to promote health. In this review, we summarize the recent advances in the nutraceutical effects and mechanism of action of carotenoids from chili pepper fruits (Table 1).

## 2. Biosynthetic Pathway of Carotenoids in Chili Pepper Fruits

The biosynthesis of carotenoids is a conserved pathway in most plant species. Precursors of carotenoids are produced through the plastidial 2-*C*-methyl-d-erythritol 4-phosphate (MEP) pathway, which leads to the formation of a molecule of isopentenyl pyrophosphate (IPP), the precursor for all isoprenoids, and its isomer dimethylallyl diphosphate (DMAPP), from pyruvate and glyceraldehyde 3-phosphate. The condensation of three molecules of IPP and one molecule of DMAPP results in the formation of geranylgeranyl pyrophosphate (GGPP), a precursor not only for the biosynthesis of carotenoids but also for chlorophyll, gibberellin, phylloquinone and tocopherol production. The first step in carotenoid biosynthesis per se is the condensation of two molecules of GGPP, catalyzed by the enzyme phytoene synthase (PSY), to produce phytoene. Successively, steps of isomerization and desaturation are carried out by the phytoene desaturase (PDS), carotene desaturase (ZDS), carotene isomerase (Z-ISO), and carotenoid isomerase (CRTISO) enzymes to give rise to the formation of all-*trans*-lycopene (Figure 2). Then, the carotenogenic pathway separates into two branches: all-*trans*-lycopene can undergo cyclization by ε-lycopene cyclase (LCY-E) to produce β-carotene or can be the substrate of the β-lycopene cyclase (LCY-B) enzyme to form α-carotene. These carotenoids are then hydroxylated, and xanthophylls, such as lutein, zeaxanthin, antheraxanthin, and violaxanthin, among others, are produced (Figure 2). Xanthophyll formation is catalyzed by β-carotene hydroxylase (BCH) and cytochrome P450-type monooxygenases 97A and 97C (CYP97) [35,36].

The zeaxanthin epoxidase (ZEP) enzyme catalyzes the conversion of zeaxanthin into violaxanthin, while violaxanthin de-epoxidase (VDE) performs the reverse reaction by converting violaxanthin into zeaxanthin. Finally, violaxanthin is transformed into neoxanthin by the catalysis of neoxanthin synthase (NSY) [11,12,35,36]. The red chili pepper fruits have the capacity to accumulate two types of ketoxanthophylls, capsanthin and capsorubin, which are produced by the transformation of antheraxanthin into capsanthin and violaxanthin into capsorubin, respectively, by the action of the capsanthin-capsorubin synthase (CCS) enzyme [37].

It has been reported that the levels of the transcripts of genes encoding the two 3-hydroxy-3-methylglutaryl-CoA reductase isozymes (HMGR1 and HMGR2) that are involved in the first step of the isoprenoid pathway showed that they are not critical for carotenoid synthesis in chili pepper. In contrast, the genes encoding geranylgeranyl pyrophosphate synthase (*GGPS*), phytoene synthase (*PSY*), and phytoene desaturase (*PDS*) are transcribed in a sequential and coordinated manner during *Capsicum* fruit ripening [38]. Additionally, the evaluation of the expression profiles of genes encoding carotenogenic enzymes showed that they are expressed at 40 to 60 days after anthesis (DAA), which corresponds to the last stage of fruit maturation [39].

It has also been reported that the color of chili pepper fruits is not only related to the expression of the *CCS* gene, but also the specific expression profile of *PSY*, *LCYB*, *CRTZ,* and *CCS*, with different profiles resulting in the diversity of colors of *Capsicum* fruit. Single and multigene silencing of these genes results in different effects on the chili pepper fruit color [40].

Functional characterization and transcript quantitation of the *LCYB*, *LCYE* and *CCS* genes in *C. annuum* var. *conoides* showed the repression of *LCYE* and the induction of the *LCYB* and *CCS* genes, demonstrating an antagonistic effect on the accumulation of carotenoids in the two branches of the carotenogenic pathway in chili pepper fruit [41]. Finally, the evaluation of xanthophyll content in four *Capsicum* varieties at 50 DAA showed significant differences in the expression profile of the genes downstream of the carotenoid pathway [42].

An interaction was reported between geranylgeranyl diphosphate synthase (GGPPS) and the small subunit homolog protein (SSUII), a small subunit protein that shares sequence similarities with GGPPS in *Capsicum annuum* var. *conoides*. This interaction enhances GGPPS activity. Additionally, a protein-protein interaction between SSUII and PSY was described, and silencing the *SSUII* gene resulted in repression of the carotenogenic pathway. These findings suggest an essential role of the CaGGPPS1/CaSSUII interaction in the regulation of carotenoid biosynthesis in chili pepper fruits [43]. On the other hand, exceptionally low levels of the transcripts of the *PSY* and *PDS* genes were detected when carotenoid biosynthesis was carried out in darkness. Nevertheless, under the same conditions, only downregulation of the *ZDS* and *LCYB* genes was reported, suggesting that the conversion or degradation of carotenoids and xanthophylls is a minor process in the dark [44].

Carotenoids in plant cells are stored in chromoplasts. They are plastids that have lost photosynthetic activity due to the absence of the chlorophyll biosynthesis machinery and the presence of proteins that participate in chlorophyll degradation [45]. Chromoplasts are specialized storage organelles capable of accumulating high levels of lipids, carbohydrates, and colorful pigments in plant tissues and organs, mainly in flowers and fruits [46]. Inside the chromoplast, carotenoids are sequestered in the membrane, fibril, crystal, tubule structures and mostly in plastoglobules [47], where the formation and organization of carotenoids occur [45]. Most enzymes of the carotenoid biosynthesis pathway, with the exception of Z-ISO, have been identified in the proteomes of chili pepper chromoplasts [48]. The carotenogenic enzymes PSY2, CRTISO and NCED are located both in the stroma and the membrane of the chromoplasts. The ZDS, LCYB, and BCH enzymes are in the plastoglobules and the chromoplast membranes. PDS and ZEP have only been found in the chromoplast membranes [48,49,50]. Finally, significant levels of *CCS* gene expression were detected in the chloroplast membranes of *C. annumm* L. fruits, and the CCS enzyme was purified and characterized [51].

## 3. Carotenoid Types and Contents in Chili Pepper Fruits

*Capsicum* spp. fruits are abundant reservoirs of carotenoids such as lutein, β-carotene, β-cryptoxanthin, zeaxanthin, violaxanthin, capsanthin and capsorubin. Carotenoid accumulation profiles change depending on the cultivar, stage of ripening and fruit color [6]. Carotenoid quantification in fruits from five cultivars of *C. annumm* L. revealed that the levels of chloroplastic pigments (lutein and neoxanthin) decreased during fruit ripening and eventually almost disappeared. Additionally, the levels of antheraxanthin, β-carotene and violaxanthin increased while pigments such as β-criptoxhanthin, zeaxanthin, capsanthin-5,6-epoxide, and cucurbixanthin A were produced de novo [52]. Moreover, a study of Peruvian chili peppers showed that β-carotene was the most abundant carotenoid in almost all samples, while capsanthin was identified only in cultivars with red fruits, and only a low content of carotenoids was detected in yellow fruits [10]. An analysis of allelic variations conducted in *Capsicum* spp. suggested that distinct combinations of dysfunctional mutations in the *PSY* and *CCS* genes could lead to different compositions and amounts of carotenoids [53].

To elucidate the causes of the differential accumulation of carotenoids in chili pepper, several investigations have been conducted. Using fruits from contrasting *Capsicum annuum* variants, it was determined that there was no positive association between the accumulation of specific carotenoids and the shape of the chromoplasts. Instead, a positive correlation between the increase in β-carotene and violaxanthin contents and an increase in total carotenoid accumulation was observed. As mentioned before, the expression levels of the *PSY* and β-carotene hydroxylase (BCH or *CrtZ-2)* genes were positively correlated with the increase in the accumulation of specific carotenoids. Additionally, no association between the transcript level of *CCS* and the carotenoid content was observed. Finally, no positive correlation between the transcript levels of the fibrillin (which encodes an important chromoplastid structural protein involved in carotenoid retention) gene and the accumulation of capsanthin was noted. However, a positive correlation between the expression of the fibrillin gene and the violaxanthin content was observed [54]. Recent advances have demonstrated that, in *C. annuum,* there are different stages of the biosynthesis and accumulation of carotenoids during fruit development, and this accumulation is associated with the esterification of xanthophylls, the expression of the acyl transferase genes and the increase in the content of fibrils inside the chromoplasts [55].

The fruit color of *Capsicum* spp. is determined by the accumulation of specific carotenoids leading to red, brown, orange, salmon and yellow fruits [56,57]. Red chili pepper fruits accumulate the six major pigments capsanthin, β-cryptoxanthin, β-carotene, capsorubin, zeaxanthin, and antheraxanthin at variable amounts and lutein at minor amounts. A high percentage of capsanthin and capsorubin is present as fatty acid esters (monoesters and diesters) [56,58]. In red chili pepper fruits, the levels of β-carotene, β-cryptoxanthin, and zeaxanthin are very low compared with capsanthin [40]. Quantitation of 34 carotenoids analyzed by HPLC in red fruits of *C. annuum* var. *lycopersiciforme rubrum* recorded 37% capsanthin, 8% zeaxanthin, 7% cucurbitaxanthin, 3.2% capsorubin and 9% β-carotene 9%. Capsanthin 5,6-epoxide, capsanthin 3,6-epoxide, 5,6-diepikarpoxanthin, violaxanthin, antheraxanthin, cryptoxanthin, several *cis* isomers, and furanoid oxides were detected in lower quantities [59]. In contrast, Mini Goggal Red, a new cultivar of *C. annuum* L., accumulated high amounts of zeaxanthin even though its apparent color was red [60].

Thus, the intensity of the red color in chili pepper fruits is determined by the content of capsanthin and capsanthin esters. Additionally, it seems that their progenitors can play key roles in the color intensity of fruits. Moreover, subplastid fractionation demonstrated that there was a differential accumulation of pigments in lines with high and low color intensity, and PSY was the most active enzyme in the membranes of plastoglobules but was inactive in the fibril fraction [55].

The brown color of the fruits of *Capsicum* spp. is due to a combination of the typical red carotenoid pigments and chlorophyll B. Lutein has also been identified in brown peppers. The brown fruits of *C. chinense* (accession AC2212) contain a high level of chlorophyll B in combination with lutein, β-carotene, β-cryptoxanthin, zeaxanthin, antheraxanthin, capsanthin, and violaxanthin, and low levels of capsorubin [56].

On the other hand, the orange color of *Capsicum* fruits is due to the accumulation of red and yellow carotenoids or to the accumulation of β-carotene (orange) [61]. For instance, fruits of some varieties of chili pepper are good sources of zeaxanthin (orange) [60]. In contrast, there are orange fruits that do not accumulate any detectable levels of carotenoids located upstream in the carotenogenic pathway, such as zeaxanthin or β-carotene [56]. Orange fruits of *C. chinense* (accession RU 72-241) were shown to accumulate low levels of capsanthin, β-cryptoxanthin and antheraxanthin. Other *Capsicum* accessions with orange fruits have been found to accumulate violaxanthin and traces of β-carotene in combination with red pigments [56].

It has been reported that distinct alleles encoding carotenogenic enzymes are related to the specific profile of carotenoids in orange chili peppers. In fact, seven different alleles of the *CCS* gene encoding at least three variants of the enzyme have been identified in *Capsicum* spp [61,62]. Other cultivars of *C. annuum* with orange fruits have been analyzed. The chili pepper ‘Fogo’ cultivar, carrying the mutation *ccs-3*, is capable of producing the corresponding transcript, but no synthesis of the functional CCS enzyme was observed, and consequently, the formation of capsanthin and capsorubin was not possible. In cultivars “Orange Grande” and “Oriole”, no transcripts of *CCS* were detected, and no red pigments accumulated. Finally, in the Canary cultivar, transcripts of the *PSY*, *LCYB*, *Crt-Z,* and *CCS* genes were detected. Nevertheless, the CCS protein did not accumulate, and red pigments were absent [63]. An induced mutation in the gene *CHY2*, encoding β-carotene hydroxylase 2, resulted in the accumulation of β-carotene as the main pigment and the conversion of the fruits from red to orange [64].

In the salmon color fruits of *C. chinense* accession RU-72-194, only traces of esters of capsanthin, capsorubin, and α- and β-carotene were identified [56]. In the Bibas accession of *C. annuum*, whose fruits are yellow, there are no detectable levels of capsanthin and capsorubin. Instead, high levels of violaxanthin and its fatty acid esters and lutein were found. Chili pepper yellow fruits accumulate α-carotene, β-carotene, zeaxanthin and antheraxanthin [56,65].

A deletion of the *PSY1* gene was identified in *C. annuum* ‘MicroPep Yellow’. In this case, the nonfunctional *PSY1* gene was compensated by the *PSY2* gene, leading to the yellow color of the fruits [66]. In the same way, deletion of the *CCS* gene is not always responsible for the yellow color of chili pepper fruits. However, different structural mutations have been identified in this gene from yellow fruits. Moreover, analysis of the promoter of the *CCS* gene suggests a complex transcriptional regulation of this gene in yellow chili pepper fruits [67,68].

## 4. Effects of Processing on the Carotenoid Content

Different methods of carotenoid extraction from plants and agro-industrial products have been performed. These methods comprise the use of solvents, enzyme-based extraction, supercritical fluid extraction, microwave-assisted extraction, Soxhlet extraction, ultrasonic extraction, and saponification. Supercritical carbon dioxide and enzyme-based extraction or the combination of two or more methods resulted in the best option to obtain high yields of carotenoids. Nevertheless, carotenoids are prone to degradation and isomerization because of the heat, light, and oxygen effects. It has been reported that capsanthin, capsorubin, and their esters are degraded at the same rate, while zeaxanthin esters respond differently to the oxidation process [69,70]. With the aim of improving the carotenoid quantity and quality, different processing techniques have been implemented and discussed.

In this sense, thermal, nonthermal, and mechanical processing have been implemented for extracting natural antioxidants from fruits and vegetables. Thermal methods such as roasting, bleaching, boiling, drying and pasteurization have been found to cause a disadvantageous effect on bioactive compounds, principally those containing pigments. However, they have been shown to protect their antioxidant effect. Nonthermal techniques such as irradiation, UV treatment, high hydrostatic pressure (HHP), and pulsed electric field (PEF) have been useful to preserve bioactive compounds, and sometimes, they have been observed to improve their antioxidant effect. In contrast, the mechanical options (chopping, trimming, peeling, crushing, slicing, sieving and pressing) have shown negative effects on the amounts, readiness and antioxidant effect of the bioactive compounds [71].

Regarding the effects of carotenoid processing, it has been reported that postharvest storage of chili pepper fruits at room temperature resulted in a gradual decrease up to 20% of the carotenoid content after one year [72]. Moreover, rapid accumulation of large amounts of phytochemicals and an attractive and intense red color was achieved when *Capsicum* fruits were incubated at 30 °C [73]. In concordance with that, low temperatures were found to decrease the content of free carotenoids by up to 66% after one year of storage [72]. In contrast, the combination of refrigeration and blue and UV-C stimuli was observed to promote the biosynthesis of chlorophylls and carotenoids without apparent damage to Habanero chili pepper (*C. chinense*) fruits during the first days of storage [74]. According to this report, red LED can accelerate the color formation and the accumulation of β-carotene, free-capsanthin, and other carotenoids in chili pepper fruits. Genes encoding the carotenogenic enzymes *LCYB*, *CrtZ,* and *CCS* have been reported to be upregulated by red LED, while blue LED stimulated the expression of the *PSY* gene. Apparently, the effect of different wavelengths of light on the accumulation of bioactive compounds is a specific characteristic of cultivars of *C. annuum* L. [75].

Cooking is a common process applied to fruits and vegetables that could change their content of bioactive compounds. Losses of 26% of β-carotene have been calculated during traditional preparation and cooking, and it should be noted that most of the carotenoids could be preserved during this process [76]. The effect of cooking on the concentration of bioactive compounds depends on the vegetable and fruit species and on the method of cooking. For instance, frying has been reported to decrease the content of polyphenols, flavonoids, carotenoids, and their antioxidant activity. Boiling and steaming have shown a variable impact on the stability of the same compounds depending on the fruit type. In chili pepper fruits, the frying process has been shown not to affect their carotenoid contents [77,78]. In general, it has been observed that boiling and, mostly, freezing processes decreased the carotenoid content in chili pepper fruits. Nevertheless, in some cases, high contents of β-carotene, β-cryptoxanthin and capsanthin after the same processing have been reported [79,80]. Finally, the effect of pasteurization on carotenoid content has been evaluated, and this process had a negative effect on the diversity of carotenoids contained in fruit juices [81].

The effect of nonthermal processing on bioactive compounds has also been evaluated in *Capsicum* spp. fruits. The drying process conserves the levels of carotenoids as well as their bioactivities [78]. Other works have concluded that milling and drying processing of chili pepper fruits decreases the content of free and esterified carotenoids up to 40% [82,83]. These contrasting results can be generated because the drying process can be influenced by factors such as the species, maturation stage, particle size of the raw material, drying temperature, and enzymatic activities, among others. In general, the traditional drying technique generates the best results with minimal carotenoid losses. Specifically, lower temperatures during drying can increase the concentration of some carotenoids in chili pepper fruits. Additionally, it has been demonstrated that the natural convective drying process can increase the content of violaxanthin in red chili pepper fruits [84,85].

The HHP process has been found to produce a positive effect on the conservation of the antioxidant activity of carotenoids, and mixing this method with thermal treatments has resulted in an increase in carotenoid bioaccessibility and antioxidant activity [86,87].

With the purpose of improving the stability of carotenoids, methods for encapsulation have been developed. Encapsulation agents such as maltodextrins, spraying, and freeze-drying are commonly used [69,88].

## 5. Carotenoid Fate after Ingestion

After ingestion, xanthophylls and carotenes must be liberated from the food matrix and dissolved into the dietary lipids and then dispersed in the digestive fluid. Subsequently, carotenoids are transferred as an emulsion to the bile salt mixed micelles. These mixed micelles are composed of free fatty acids, lyso-phospholipids, mono- and diacylglycerides, and free cholesterol that are produced during the digestion of triglycerides, phospholipids, and cholesterol esters. The solubilization of carotenoids in the mixed micelles enables their incorporation into intestinal epithelial cells. Once inside the cells, the fatty acids and monoacylglycerols are re-esterified into triglycerides and assembled to form chylomicrons, secreted across the basolateral membrane into the lymph and then transported for circulating in the bloodstream [89,90,91].

Later, the lipoprotein lipase degrades the chylomicrons, and their remnants are taken up by liver hepatocytes. Inside the liver, carotenoids are liberated from the remnant and secreted as very low-density lipoprotein (VLDL) and transported into the plasma. VLDLs are then transformed into low-density lipoprotein (LDL) with less triglyceride content. Finally, the carotenoids are incorporated into tissues after uptake by the LDL receptor. β-carotene and lycopene are delivered by LDL particles, whereas xanthophylls such as lutein and zeaxanthin are associated with high-density lipoprotein (LDH). Inside the cells, provitamin A carotenoids can be transformed into retinal due to the action of β-carotene 15,15′-dioxygenase (BCO1). Then, the retinal is reduced and converted into retinol by the catalysis of retinol dehydrogenase (RDH), and finally, retinol is esterified by the action of lecithin-retinol acyltransferase (LRAT)-producing retinyl esters. On the other hand, the enzyme BCO2 is involved in the cleavage of provitamin A and nonprovitamin A carotenoids, generating β-apo-carotenoids [89,90,91].

The presence of carotenes and xanthophylls from chili pepper fruits after ingestion was evaluated. Capsanthin is metabolized within ten hours after ingestion and converted into capsanthone [92]. Increases of 1.2- and 2.2-fold levels of carotenoids (capsanthin, capsanthone, cucurbitaxanthin A and cryptocapsin) were detected in human plasma and erythrocytes, respectively, after two weeks of regular ingestion [93]. Moreover, supplementation with an oleoresin extracted from *C. annuum* L. containing a mixture of carotenoids (capsanthin, β-cryptoxanthin, zeaxanthin, and β-carotene) increased the accumulation of β-cryptoxanthin in plasma more than that of other carotenoids after 12 weeks of ingestion without adverse effects on health [94].

The bioaccessibility (the ratio of carotenoids solubilized in micelles to the total carotenoids ingested) of carotenoids and xanthophylls in chili pepper fruits depends on the species, cultivar, and fruit color. For instance, the bioaccessibility of carotenoids from yellow peppers has been observed to be higher than that of red chili peppers [95]. In contrast, some other reports have suggested that red chili peppers have a higher content and bioaccessibility than orange and yellow peppers [96]. It has been reported that xanthophylls exhibit greater bioaccessibility than carotenoids because they are more successfully transferred to micelles [80]. In raw chili pepper fruits, capsanthin and zeaxanthin have the highest bioaccessibility, followed by antheraxanthin. β-cryptoxanthin, violaxanthin, and β-carotene, whereas neoxanthin and lutein were not detected in the micelles [79].

Some reports have suggested that the cooking process could increase food bioaccessibility. For example, in *Capsicum* fruits, freezing increased the bioaccessibility of capsanthin, zeaxanthin, antheraxanthin, β-cryptoxanthin, and violaxanthin, whereas boiling decreased the bioaccessibility of capsanthin and zeaxanthin [79,80,85].

Bioavailability is defined as the percent of nutrients ingested that are absorbed in the small intestine. Thus, the bioavailability of carotenoids depends on its bioaccessibility [91]. It has been reported that some processing methods such as heating can improve the bioavailability of bioactive compounds of chili pepper [85]. Moreover, absorption of carotenoids can be increased by the addition of excipient ingredients such as oils, or by incorporating different emulsifiers in foods [91,97].

## 6. Carotenoids as Antioxidants

Carotenoids are powerful antioxidant agents. They act on a wide range of oxidizing radicals through the electron transfer process. Their antioxidant properties are usually associated with the capacity to remove free radicals and single oxygen. The interaction of carotenoids with reactive oxygen species (ROS) can be carried out through oxidation, reduction, hydrogen atom abstraction or additional reactions. Additionally, the structure of carotenoids allows its incorporation inside the hydrophobic membranes parallel to the membrane surface (as β-carotene or lycopene) or as zeaxanthin and astaxanthin, whose polar hydroxyl groups lead to anchoring to the membranes with polar functional groups oriented outside the membrane, preventing the entry of oxygen inside the membranes. Finally, it has been reported that the antioxidant properties of carotenoids could depend on the presence of other co-antioxidants, such as tocopherol. α-Tocopherol, β-carotene, and ascorbic acid act synergistically in the carotenoid regeneration process to avoid any pro-oxidant behavior (at high concentrations of oxygen, carotenoids might have a pro-oxidant effect instead of antioxidant activity) [98,99].

ROS and radicals derived from nitrogen can cause oxidative stress or nitrosative stress, respectively, in biological systems. This negative impact includes damage to biomolecules, oxidation of lipids, membrane alterations, peroxidation of fatty acids and even fragmentation and reactivity damage to DNA. In fact, many degenerative disorders such as cancer, cardiovascular disease, stroke, cataract, degeneration of the macular region of the retina, immunosenescense, and aging are associated with oxidative damage [98,99,100]. Because of the biological importance of carotenoids, studies on the carotenoid content associated with their antioxidant properties have been carried out in chili pepper fruits. In general, the high content of capsanthin and other carotenoids, ascorbic acid, and capsaicin has been associated with the antioxidant properties of *Capsicum* spp. [101]. Additionally, the antioxidant power of chili pepper fruits can be influenced by environmental conditions and the harvesting stage [102,103]. Few specific analyses of chili pepper carotenoids and their antioxidant potential have been reported. Evaluation of extracts of *Capsicum annuum* L. cv. Senise from the Basilicata region containing β-carotene, capsorubin, antheraxanthin, and β-cryptoxanthin revealed antioxidant activity (quenching peroxyl radicals in a dose-dependent manner). Moreover, the transcription of genes involved in the redox balance were activated by the treatment with these chili pepper extracts. Furthermore, an increase of the antioxidant effect was reported when extracts of chili pepper fruits were incorporated into liposomes as nanocarriers [104].

In epithelial cells from the rat liver, a red chili pepper extract from *C. annuum* L., capsanthin and β-carotene showed a protective effect on the inhibition of gap-junction intercellular communication induced by hydrogen peroxide. Additionally, carotenoids suppressed the formation of ROS in the same cells treated with hydrogen peroxide. These results suggest that the dietary intake of red paprika containing capsanthin and β-carotene may help lower the risk of deteriorative diseases caused by oxidative stress in the human body [105].

It is possible to analyze the antioxidant activity of capsanthin and its esters. Hydroperoxide formation is suppressed by capsanthin, β-carotene, lutein, and zeaxanthin via free radical oxidation. Capsanthin has a slower degradation rate than the other carotenoids, and thus, its radical scavenging effect is prolonged. The radical scavenging ability of capsanthin was not influenced by the esterification process. Capsanthins, monoesterified or diesterified, were also good radical scavengers [106].

Capsanthin and capsanthin 3,6-epoxide exhibit notable superoxide generation and inhibitory effects on the generation of nitric oxide (NO), whereas lutein, astaxanthin and canthaxanthin significantly increased NO generation. It has been demonstrated that the structure of pigments can play a key role in their functions. The 3-hydroxy-κ-end groups of capsanthin and capsorubin can suppressed NO generation, whereas the 4-exo-β-end group of the other carotenoids enhanced NO generation [107,108].

Because carotenoids are capable of inactivating free radicals and modulate gene expression, dietary intake of these pigments have beneficial effects on health. For instance, carotenoids have a hepatoprotective effect by reducing oxidative stress and regulating the lipid metabolism of hepatocytes and, thus, leading to a reduction of the risk of developing liver diseases such as non-alcoholic fatty liver disease (NAFLD), among others [109]. The antioxidant effect of carotenoids concerning other diseases and health benefits are discussed in the following sections of this article.

## 7. Carotenoids and Cancer

Pigments, such as β-cryptoxanthin, neoxanthin, zeaxanthin capsanthin, capsorubin, and lutein, have been analyzed because of their potential role in cancer treatment and prevention [110]. Several studies have determined that these pigments are capable of attenuating oncogene signaling, triggering apoptosis of cancer cells, regulating cell cycle progression, dynamically modulating the redox balance, inhibiting tumor-specific angiogenesis, controlling tissue invasion and metastasis, modulating gap junction intercellular communications, and modulating multidrug resistance [111,112].

The cytotoxic, genotoxic and antiproliferative effects of carotenoids in tumoral cells can be explained through diverse biological processes. Carotenoids can cause a pro-oxidative effect exclusively on cancer cells, generating and increasing the levels of ROS accumulation as a key method of selectively killing the cancer cells [112].

The mechanisms to suppress cancer progression can include reductions in the activity and phosphorylation of mitogen-activated protein kinases (MAPKs), such as extracellular signal-regulated kinase (ERK) and c-Jun N-terminal kinases (JNK), and inhibition of the signaling route of phosphatidylinositide 3-kinase (PI3K)/protein kinase B (Akt)/mammalian target of rapamycin (mTOR). Another mechanism is inhibition of the pro-apoptotic proteins B-cell lymphoma (Bcl-2) homologous antagonistic/killer 1 (Bak1), Bcl-2 associated death promoter (Bad), and Bcl2 associated X protein (Bax). Additionally, activation of the caspase cascade enhances apoptosis [21,111].

Carotenoids can also enhance the activity of tumor suppressor proteins: retinoid x receptor (RXR), peroxisome proliferator-activated receptor gamma (PPARγ), peroxisome proliferator-activated receptor (PRAR), p21, p27, and p53, among others). They can reduce the activity of cancer inductors nuclear factor kappa B (NF-κB), X-linked inhibitor of apoptosis protein (XIAP), survivin, matrix metalloproteinases (MMPs), S-phase kinase-associated protein 2 (Skp2), urokinase plasminogen activator (uPA), cell surface glycoprotein (CD44), chemokine receptor (CXCR4), and hypoxia-inducible factor-1α (HIF-1α) [111].

Finally, they can arrest the cell cycle by suppressing the activation of the cyclin CDK complex, suppress metastasis through enhanced gap junction intercellular communication and E-cadherin, reduce angiogenesis by suppressing the activities of vascular endothelial growth factors, and reverse multidrug resistance by the inhibition of ABC transporters [111,113].

The presence of certain plasma carotenoids has been inversely related to the occurrence of PSA. Thus, α-, β-carotene and β-cryptoxanthin have been widely studied for the prevention of aggressive prostate cancer [20,21,114]. Additionally, an inverse association between total dietary carotenoids (α-, β-carotene, β-cryptoxanthin, lutein, and zeaxanthin) and the risk of gastric cancer has been reported [15]. Moreover, β-carotene, lutein, zeaxanthin, and β-cryptoxanthin have been demonstrated to have a preventive effect on breast and lung cancer [21].

In particular, β-cryptoxanthin seems to be a promising pigment for cancer prevention. Its proposed mechanisms of action are through diverse signaling pathways, such as NF-κB, RAR, PPARs, sirtuin (SIRT) factors, and the nicotinic acetylcholine receptor. Effects on the tumor suppression proteins p53 and p73 and modulation of the reverse cholesterol transport mechanism have also been considered as possible mechanisms [24].

In addition, the activities of capsanthin and capsorubin for cancer prevention have been studied. An inhibitory effect of capsanthin on colon carcinogenesis has been reported [27]. Moreover, capsanthin, capsanthin 3′-ester, and capsanthin 3,3′-diester have shown in vitro antitumor promoting activity on mouse skin two-stage carcinogenesis [28]. Furthermore, the modulation of multidrug resistance and apoptosis of cancer cells was strongly promoted by capsanthin and capsorubin and moderately promoted by lutein and violaxanthin [16].

More recent reports have determined that the application of capsanthin on MCF-7 human breast cancer cells decreased their viability within 24 h of its application. Additionally, in a dose-dependent manner, capsanthin caused oxidative stress and DNA damage. An increase in the mitochondrial apoptotic mechanism associated with an increase in the levels of p53 and Bax protein and a decrease in Bcl2 protein was also observed. Finally, treatment with capsanthin caused an increase in the levels of catalase and glutathione, leading to an increase in lipid peroxidation [29].

Although in vitro and in vivo studies about the chemoprotective action of carotenoids have shown promissory advances, clinical trials have shown either positive, negative, or non-effect of carotenoids in cancer prevention or treatment [113].8. Carotenoids and Cardiovascular Disorders.

Numerous studies have been conducted to analyze the association between the intake of carotenoids such as α- and β-carotene, β-cryptoxanthin, and zeaxanthin and cardiovascular health and atherosclerosis. In fact, the effect of β-carotene on heart diseases has been evaluated through some clinical trials [115,116].

Because carotenoids have antioxidant and anti-inflammatory activities, they might help improve cardiovascular health through the maintenance of blood pressure baseline levels, the reduction of pro-inflammatory cytokines and markers of inflammation (such as C-reactive protein), and correction of dyslipidemia [117].

Atherosclerosis represents an important risk factor for cardiovascular diseases. In this sense, it has been reported that the antioxidant and anti-inflammatory effects of lutein in aortic tissues could protect against the development of atherosclerosis in animal and human models [118,119]. Moreover, a high intake of lutein and high levels of lutein in the blood have been correlated with better cardiometabolic health performance [120]. Further, a study carried out on young adults demonstrated an association between high levels of circulating carotenoids (β-carotene, lutein, zeaxanthin and β-cryptoxanthin) and a low risk of future hypertension problems [121].

A significant increase was noted in the levels of HDL cholesterol in the plasma of young male Wistar rats fed purified capsanthin for two weeks. Analysis of the hepatic transcripts showed that capsanthin could upregulate the mRNA levels of the hepatic lipase ApoA5 and the lecithin cholesterol acetyltransferase genes. Thus, capsanthin could lead to an increase in cholesterol efflux to HDL particles by increasing the levels of Apo5 and/or by enhancing lecithin cholesterol transferase activity [30].

## 8. Carotenoids as Anti-Inflammatory Agents

Chronic inflammation is associated with several diseases. The production of ROS (singlet oxygen and hydrogen peroxide) is characteristic in sites of chronic inflammation. In this sense, intake of dietary antioxidants such as carotenoids has been suggested as an alternative to prevent inflammatory responses [122]. The anti-inflammatory effects of carotenoids involve the suppression of NF-κB activity, the downregulation of pro-inflammatory molecules, the protection of membranes against oxidative damage, and the promotion of the activity of antioxidant enzymes [123]. The anti-inflammatory effect of chili pepper fruits has been evaluated, and it has been reported that a carotenoid extract from guajillo chili pepper containing violaxanthin, β-cryptoxanthin and β-carotene showed analgesic activity, probably produced by inhibition of the local prostaglandins. Furthermore, high doses of a carotenoid extract were capable of significantly inhibiting the formation and progression of edema in mice [22].

The anti-inflammatory properties of an extract from Ukrainian Cayenne chili pepper fruits consisting mainly of capsanthin, lutein and β-carotene were evaluated in an adjuvant-induced inflammation model in rats. A linear up to 50% decrease in edema around the area of inflammation was induced by the carotenoid extract. Additionally, a 1.3-fold reduction in the levels of acetylcholinesterase activity and a two-fold decrease in the concentration of seromucoids in the serum of rats were recorded [17].

Rats fed a high-fat diet exhibited increased oxidative stress and upregulation of inflammatory indicators. Supplementation of the diet with lutein modulated the expression of genes involved in oxidative stress and inflammation, including NF-κB and the nuclear factor erythroid 2 (Nrf2) signaling pathways in the retina, which might contribute to ameliorating the retinal damage induced by the high-fat diet [18].

Lutein has been shown to exert anti-inflammatory effects on endotoxin-induced uveitis (EIU) in a dose-dependent manner in rats. A reduction in the levels of NO, tumor necrosis factor (TNF-α), interleukin (IL-6), prostaglandin (PGE2), monocyte chemoattractant protein (MCP 1), and macrophage inflammatory protein (MIP 2) was reported. Moreover, lutein inhibited the NF-κB-dependent signaling pathway and ameliorated the subsequent production of pro-inflammatory reactions [124].

A study carried out on 45- to 75-year-old persons revealed that β-cryptoxanthin intake has been associated with low risks of developing inflammatory disorders such as rheumatoid arthritis [125]. β-carotene has been shown to reduce the inflammatory response in endothelial cells from the human umbilical vein. β-Carotene was shown to modulate the pro-inflammatory factor TNF-α, associated with a reduction in the generation of ROS and nitrotyrosine, an increase in nitric oxide/cyclic guanosine monophosphate levels, and downregulation of the expression levels of *NF-κB* genes [126]. Carotenoids have shown not only free radical scavenging and quenching activities but have also been found to be capable of suppressing singlet oxygen and NO generation in inflammatory leukocytes, as well as in neutrophils and macrophages [107].

Carotenoids influence the immune system (cellular and humoral) because it is extremely sensitive to the oxidative stress. It seems that carotenoids participate in the maintenance of the balance between the elimination of active oxygen species and the optimum cell signaling (reducing the ROS damage to the cell membranes and their receptors). Carotenoids also influence the production of cytokines and prostaglandins and the activity of the redox-sensitive transcription factors [100,127].

The research about the role of carotenoids on immunity includes immunoglobulin (Ig) production, lymphoblastogenesis, lymphocyte cytotoxic activity, cytokine production, and the delayed-type hypersensitivity (DTH) [128].

Retinoic acid (produced through β-carotene conversion) is a relevant immunomodulatory molecule. Thus, β-carotene increases the content of CD3+, CD4+, and CD8+ lymphocytes in the spleen, and increase DHT and natural killer (NK) cells activity [100,129].

In this sense, it has been established that β-cryptoxanthin stimulates the humoral immunity in mammals. Using rabbits as model, the administration of β-cryptoxanthin leads an increase in the number of CD4+ lymphocytes in the blood, and also to an elevation of the serum levels of IgG, IgM, IgA, and IL-4 [130].

## 9. Carotenoids and Obesity

Several works have supported the anti-obesity effect of carotenoids because they have been involved in the modulation of key aspects of adipose tissue biology. The anti-adiposity and anti-inflammatory mechanisms of action of carotenoids seem to be associated with interactions with transcription factors of the nuclear receptor superfamily and modulation of the pro-inflammatory and antioxidant signaling pathways in adipocytes and cells of the stromal-vascular fraction [131]. Carotenoids have been reported to regulate gene expression in adipocytes and adipose tissue. Provitamin A carotenoids leading to retinoid acid synthesis and retinoid X receptors were demonstrated to regulate gene expression via ubiquitous signaling pathways, such as NF-κB and MAPKs [25].

Treatment with pigments from chili pepper fruits has been observed to suppress the chronic inflammation caused by obesity in 3T3-L1 adipocyte cells. These compounds could promote 3T3-L1 differentiation and upregulation of the expression of the adiponectin gene and increased secretion of its protein. Coculture of adipocyte and macrophage cells treated with chili pepper pigments suppressed the expression of the IL-6, TNF-α, MCP-1, and resistin genes. Additionally, chili pepper pigments can affect adipocytokine secretion and might, therefore, affect antimetabolic syndrome diseases [132].

Research into correlations between the intake of xanthophyll capsules from red chili pepper fruits and diminution of the abdominal fat area have been performed. The results showed a significant reduction in the abdominal fat area and a decrease in the body mass index in healthy volunteers without adverse effects after the intake of red pepper xanthophylls for 12 weeks. This report suggested that *Capsicum* xanthophylls were safe compounds that improved lipid metabolism and reduced abdominal fat [133]. Moreover, the effect of red chili pepper and capsanthin on lipid metabolism was evaluated in obese mice, and a reduction in weight gain and amelioration of the hypertrophy of the adipose tissues and liver were note. Additionally, an improvement of the serum lipid profile and adipokine secretion, an amelioration of hepatic steatosis due to a suppression of hepatic lipogenesis and gluconeogenesis, and an increase in fatty acid oxidation were reported after treatment for eight weeks. Finally, red pepper and capsanthin were capable of inhibiting adipogenesis and reducing lipid droplet sizes [31].

In addition, using a murine preadipocyte cell line (3T3L1), it was possible to establish that capsanthin had both anti-obesity and insulin-sensitizing activities. This red pigment showed lipolytic activity on differentiated adipocytes and was able to inhibit adipogenesis. Additionally, in high-fat diet-fed animals, capsanthin significantly increased their locomotive activity and therefore caused weight loss generated by excessive ATP production triggered by an increase in lipolytic activity and accelerated oxidation of fatty acids due to the adrenoreceptor β_2_-agonistic effect of capsanthin. Capsanthin also dose-dependently increased adiponectin and p-AMPK activity in high-fat diet-fed animals [32].

Assays carried out with C57BL/6J mice have shown a significant reduction in weight gain induced by a high-fat diet after capsanthin treatment. An important decrease in the serum levels of total cholesterol, triglycerides, and low-density lipoprotein cholesterol, and a reduction in the levels of trimethylamine N-oxide (TMAO), a detrimental metabolite for cardiovascular health associated with atherosclerosis, were detected after 12 weeks of treatment. Moreover, an increase in intestinal microbiota diversity was observed since an analysis of 16S rRNA gene sequencing showed an increase in the abundance of Bacteroidetes (*Bifidobacterium* and *Akkermansia*) and a decrease in the abundance of *Ruminococcus* and the ratio of Firmicutes/Bacteroidetes. Additionally, it has been demonstrated that capsanthin increased the abundance of replication and repair gene expression and decreased the gene expression of membrane transporters and carbohydrate metabolism [33].

Pigments such as β-carotene and β-cryptoxanthin have been widely analyzed because of their anti-obesity effects. They have been involved in reductions of body weight, adipose tissue mass, adipocyte hypertrophy, serum lipid concentrations, and the inhibition of adipogenesis [131]. β-carotene seems to decrease obesity by promoting fatty acid oxidation in adipocytes and other tissues. It has been reported that β-carotene exerts an anti-obesity effect only when it is converted into vitamin A. Based on that, experimental evidence supports that treatment with retinoic acid affects adipogenesis, promotes adipose tissue browning, and favors an overall induction of fatty acid oxidation in adipocytes [23]. Anti-adiposity properties have also been reported for β-cryptoxanthin, but the mechanism of action is still unknown, while zeaxanthin has been observed to inhibit obesity in high-fat diet-fed mice, apparently by inducing AMPK activation and by inhibiting lipogenesis in the adipose tissue [25].

## 10. Miscellaneous Effects

In addition to the issues discussed above, various other health effects of carotenoids have been reported.

Due to their ability to absorb UV radiation and because of their distinctive structure and chemical properties, susceptibility to oxidation, rigidity, tendency for aggregation, and even their fluorescence properties (as phytofluene), a photoprotective effect of carotenoids has been described [134]. In this regard, the intake of a red chili pepper extract with xanthophylls and a combination of oral and topical xanthophyll treatment has shown a significant reduction in skin oxidative stress and a decrease in erythema and skin darkening, in summary, the maintenance of skin health [135]. It has been demonstrated that capsanthin, capsorubin and lutein could improve natural photoprotection. Pretreatment with these pigments was capable of counteracting the cytotoxic effects of UV radiation by decreasing the formation of DNA strand breaks and the cleavage action of caspase 3 (a marker of apoptosis produced by UV radiation) in human dermal fibroblasts [34].

Moreover, several reports have suggested positive effects of lutein and zeaxanthin in promoting visual health and reducing the risks of eye diseases such as age-related macular degeneration and cataracts. Additionally, supplementation with lutein and zeaxanthin can increase the macular pigment density and, therefore, can offer protection against ocular diseases. In this sense, the mechanism of action has been elucidated. Lutein and zeaxanthin are pigments capable of absorbing light in a broad wavelength range (400–475 nm), allowing an attenuation of high light exposure. Additionally, their conjugated double bonds and their hydroxyl substituents make it possible to establish hydrogen bonds with the polar head groups at the membrane surfaces of the eye tissues. On the other hand, their antioxidant properties can protect tissues from light oxidative damage [19]. Lutein has been found to reduce or prevent the generation of ROS under light exposure. Lutein exhibited not only ROS scavenging activity but was also capable of inducing endogenous antioxidant superoxide dismutase gene expression and activity. This activity promoted tight junction repair in the retinal pigment epithelium and reduced monocyte chemotactic protein-1 and the subsequent macrophage infiltration. Additionally, lutein promoted tight junction repair and suppressed inflammation in photostressed mice, reducing local oxidative stress by direct scavenging and most likely by the induction of endogenous antioxidant enzymes [136].

In addition, antidiabetic effects of carotenoids have also been described. In healthy men and women, a dietary intake of high levels of α- and β-carotene has been associated with a reduction in developing type 2 diabetes [137]. Consistent with this finding, a recent report suggested that capsanthin was capable of improving the tolerance to glucose in mice, while carotenoids, in general, improved insulin sensitivity in muscle, liver and adipose tissues and modulated the expression of specific genes involved in cell metabolism [33,117]. Furthermore, a transcriptomic analysis in the nematode *Caenorhabditis elegans* concluded that β-cryptoxanthin promoted the upregulation of energy metabolism, stress response, and protein homeostasis, preventing diseases related to metabolic syndrome and aging [138].

Finally, the dietary intake of lutein and β-carotene has been positively associated with memory performance. In particular, lutein seems to have an important role in hippocampal function in adults [139]. Moreover, several studies have supported that carotenoid intake might have therapeutic potential in the prevention and amelioration of neurodegenerative diseases through mechanisms such as the upregulation of antioxidant enzyme genes, ROS quenching, and anti-inflammatory activity [123,140]. In this sense, carotenoids of chili pepper fruits (capsanthin, lutein, zeaxanthin, β-cryptoxanthin, α-carotene, *trans*-carotene, and *cis-*carotene) have shown anti-Alzheimer disease properties by inhibiting key enzymes relevant to this ailment, such as acetylcholinesterase, butyrylcholinesterase, and β-secretase [26].

## 11. Conclusions

Chili pepper is a promising model as a source of bioactive compounds, mainly carotenes and xanthophylls. In fact, ripe chili pepper fruits, regardless of their color, have carotenoids that can have a beneficial effect on health. Biochemical, molecular, and cellular mechanisms of carotenoid biosynthesis and accumulation in *Capsicum* spp. are complex, and the carotenogenic pathway, which must be regulated at the transcriptional and posttranscriptional levels, remains to be elucidated. The dietary intake of fruits and vegetables rich in carotenoids, such as chili pepper fruits, has important benefits for health (Table 1). However, it is necessary to carry out future investigations for a better comprehension of the mechanisms of action of the beneficial effects of carotenes and xanthophylls of *Capsicum* spp. in different diseases or disorders and their health implications.

## Figures and Tables

**Figure 1 molecules-25-05573-f001:**
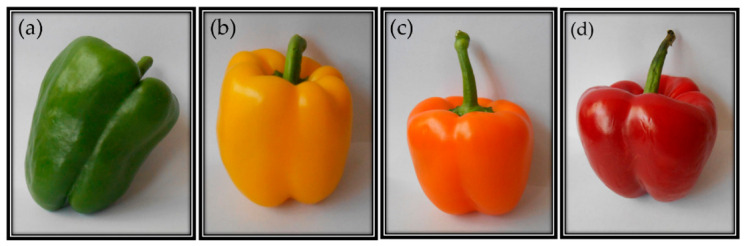
Commercial bell pepper fruits of *Capsicum* spp. showing different colors due to the presence of (**a**) chlorophylls and (**b**–**d**) carotenoids.

**Figure 2 molecules-25-05573-f002:**
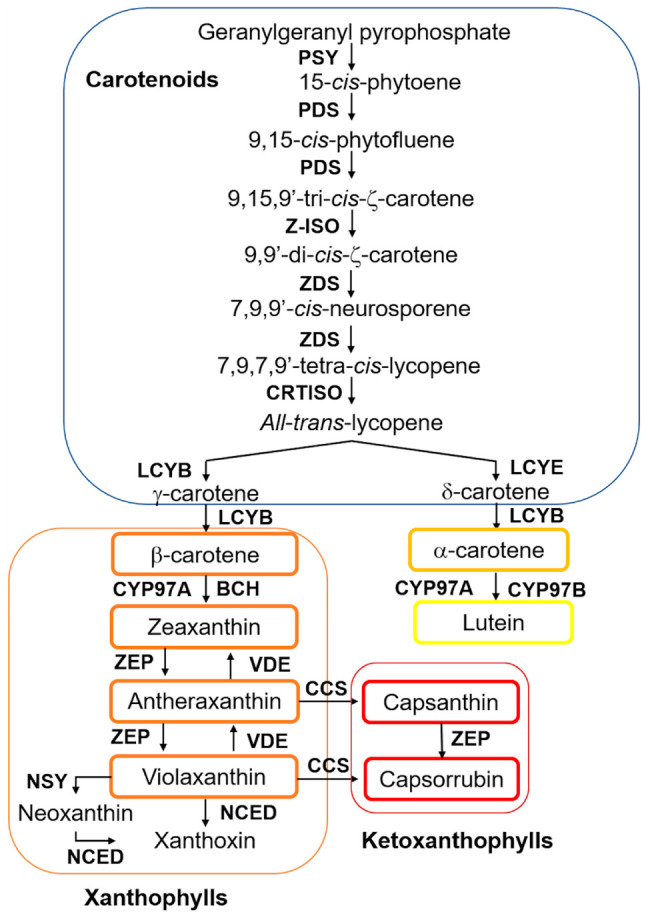
Chili pepper fruit carotenoid biosynthetic pathway. PSY (phytoene synthase), PDS (phytoene desaturase), Z-ISO (ζ-carotene isomerase), ZDS (ζ-carotene desaturase), CRTISO (carotene isomerase), LCYB (β-lycopene cyclase), LCYE (ε-lycopene cyclase), BCH (β-carotene hydroxylase), CYP (β-carotene hydroxylase cytochrome 450 type A and B), ZEP (zeaxanthin epoxidase), VDE (violaxanthin epoxidase), CCS (capsanthin-capsorubin synthase), NSY (neoxanthin synthase), NCED (9-*cis*-epoxycarotenoid dioxygenase). Modified from [35,36].

**Table 1 molecules-25-05573-t001:** Chili pepper fruit carotenoids and their nutraceutical effects.

Carotenoid	Health Effect	Mechanism of Action	Reference
Lutein	Gastric cancer	Not determined (ND)	[15]
	Cancer cells	Modulation of apoptosis and multidrug resistance	[16]
	Edema reduction	Reduction acetylcholinesterase, Increase seromucoids	[17]
	Retina damage	Modulation of oxidative stress, and pro-inflammatory gene expression	[18]
	Macular degeneration	Absorption of UV radiation, antioxidant	[19]
β-carotene	Prostate cancer	Inverse correlation with prostate-specific antigen (PSA) occurrence	[20,21]
	Gastric cancer	ND	[15]
	Anti-inflammatory, analgesic, antinociceptive	ND	[22]
	Edema reduction	Reduction acetylcholinesterase, Increase seromucoids	[17]
	Obesity	Promotion of fatty acid oxidation	[23]
β-cryptoxanthin	Prostate cancer	Inverse correlation with PSA occurrence	[20,21]
	Gastric cancer	ND	[15]
	Cancer prevention	Modulation of signaling pathways	[24]
	Anti-inflammatory, analgesic, antinociceptive	ND	[22]
Zeaxanthin	Gastric cancer	ND	[15]
	Obesity	Activation of AMP-activated protein (AMPK) and inhibition of lipogenesis	[25]
	Macular degeneration	Absorption of UV radiation, antioxidant	[19]
	Alzheimer disease	Inhibition of acetylcholinesterase, butyrylcholinesterase and β-secretase	[26]
Violaxanthin	Cancer cells	Modulation of apoptosis and multidrug resistance	[16]
	Anti-inflammatory, analgesic, antinociceptive	ND	[22]
Capsanthin	Colon cancer	Inhibitory effect	[27]
	Skin cancer	Chemopreventive	[28]
	Cancer cells	Modulation of apoptosis and multidrug resistance	[16]
	Cancer breast(MCF-7 cells)	Oxidative stress, DNA damage, increase p53 and Bax, lipid peroxidation	[29]
	Atherosclerosis	increase in the cholesterol efflux	[30]
	Edema reduction	Reduction of acetylcholinesterase, Increase seromucoids	[17]
	Obesity	Suppression of hepatic lipogenesis, fatty acid oxidation, and gluconeogenesis. Inhibit adipogenesis	[31]
	Obesity and insulin sensitizing	Inhibition of adipogenesis, increase of lipolytic activity, accelerated oxidation of fatty acids	[32]
	Atherosclerosis	Decrease on serum levels of total cholesterol, triglycerides, low density lipoprotein cholesterol, prebiotic	[33]
	Skin health	Counteract the cytotoxic effect of UV radiation by decreasing the formation of DNA strand breaks	[34]
	Diabetes	Improvement of glucose tolerance, improvement of insulin sensitivity	[33]
	Alzheimer disease	Inhibiting acetylcholinesterase, butyrylcholinesterase and β-secretase	[26]
Capsorubin	Cancer cells	Modulation of apoptosis and multidrug resistance	[16]
	Skin health	Counteract the cytotoxic effect of UV radiation by decreasing the formation of DNA strand breaks	[34]

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
