# Peer review of "Chili Pepper Carotenoids: Nutraceutical Properties and Mechanisms of Action"

_molecules, 2020, doi:10.3390/molecules25235573_

Round 1

Reviewer 1 Report

Dear Authors

The review here presented is based on the description of chili pepper carotenoid, their potential as nutraceuticals and beneficial properties.

Though the study is within the aim and scope of the journal, my suggestion is to better discuss the inhibitory effect and cytotoxicity value of Capsicuum Annum species from traditional recognized culture, such as that of Altino or Basilicata, as recently reported by diverse authors.

English needs revision for typos and mistakes.

For this reason the manuscript requires extensive revision. 

Author Response

Reviewer 1

The review here presented is based on the description of chili pepper carotenoid, their potential as nutraceuticals and beneficial properties.

Though the study is within the aim and scope of the journal, my suggestion is to better discuss the inhibitory effect and cytotoxicity value of Capsicuum Annum species from traditional recognized culture, such as that of Altino or Basilicata, as recently reported by diverse authors.

Response: Although interesting reports about the enzyme inhibitory effect, antioxidant, and anti-inflammatory activity of extracts of sweet and hot peppers from Altino (Chieti, Italy) on prostate cancer cell line 3 (PC3) were recently published, these extracts were principally composed of flavonoid, phenolic compounds, and fatty acids (See Della Valle et al., 2020. Exploring the nutraceutical potential of dried pepper Capsicum annuum L. on Market from Altino in Abruzzo Region. Antioxidants (5), 400, doi: 10.3390/antiox9050400). We would like to address that our review is focused on the nutraceutical properties of carotenoids and no on other bioactive compounds of Capsicum species.

On the other hand, the antioxidant effect of extracts of Capsicum annuum L. cv. Senise from the Basilicata region, containing carotenoids, was incorporated in section 5 “Carotenoids as Antioxidants” (Lines 360-365).

English needs revision for typos and mistakes.

Response: An English edition service (American Journal Experts) was used for the correction of this manuscript previously to the submission, and the certificate of the edition is attached.

For this reason the manuscript requires extensive revision.

Reviewer 2 Report

The manuscript reviewed various bioactivities of chili pepper carotenoids. Overall, the manuscript is well written and easy to follow. The manuscript can be acceptable for publication in molecules after proper answer for following questions or suggestions.

1) In description about 5. Carotenoid fate after ingestion

Differentiate “bioaccessibility” and “bioavailability” carefully. Please provide a little more information about “how to improve bioavailability”

2) P. 338-342, the meaning of sentence is not clear. What do you mean “ to avoid prooxidant behavior”

3) 6. Carotenoid as Antioxidants need to be more elaborated. Please provide more substantial evidences.

4) The cancer preventive effects of carotenoids are conclusive in human clinical trials?

5) Please add immune modulation effects of carotenoids in 9. Carotenoids as anti-inflammatory agents.

Author Response

Reviewer 2:

  • In description about 5. Carotenoid fate after ingestión

Differentiate “bioaccessibility” and “bioavailability” carefully. Please provide a little more information about “how to improve bioavailability”.

Response: The concept of bioavailability and information about how to improve the bioavailability of carotenoids was incorporated into the manuscript (Lines 328-332). Besides, we checked that all reports that we mentioned in this section were correctly referred to bioaccessibility.

  • 338-342, the meaning of sentence is not clear. What do you mean “to avoid prooxidant behavior”.

Response: The explanation of prooxidative behavior was added to this corrected version of the manuscript (Lines 344-346).

3) 6. Carotenoid as Antioxidants need to be more elaborated. Please provide more substantial evidences.

Response: “Carotenoids as Antioxidants” section was more elaborated in this corrected version, and some changes were carried out:

  • The sentence “It has been suggested that this oxidative stress could be the common cause of various diseases, such as cancer, diabetes or cardiovascular damage” was changed by “In fact, many degenerative disorders such as cancer, cardiovascular disease, stroke, cataract, degeneration of the macular region of the retina, immunosenescense, and aging are associated with oxidative damage” (Lines 350-353).
  • The antioxidant effect of chili pepper extracts and its incorporation into liposomes was added (Lines 360-365).
  • The hepatoprotective effect of carotenoids was incorporated (Lines 384-388).
  • The paragraph “Recent reports have indicated that the formulation of extracts of chili pepper fruits in nanocarriers as liposomes could enhance their antioxidant effects” was deleted (Lines 390-391).

As mentioned in the text (Lines 388-389), most nutraceutical properties of carotenoids are associated with their antioxidant activity, and more substantial evidence of that effect is discussed in sections 7, 8, 9, 10 and, 11 of this review.

4) The cancer preventive effects of carotenoids are conclusive in human clinical trials?.

Response: It has been reported that carotenoids act as chemoprotective molecules in several cancer types; nevertheless, clinical trials have shown contradictory results. This correction was integrated into the corrected version of this manuscript (Lines 445-447).

5) Please add immune modulation effects of carotenoids in 9. Carotenoids as anti-inflammatory agents.

Response: Immune modulation effects of carotenoids were included in “Carotenoids as anti-inflammatory agents” (Lines 508-523).

Round 2

Reviewer 1 Report

Dear Authors

The manuscript has been revised according to reviewers suggestions, now it is suitable for publication.